# Piloting an In Situ Training Program in Video Consultations in a Gynaecological Outpatient Clinic at a University Hospital: A Qualitative Study of the Healthcare Professionals’ Perspectives

**DOI:** 10.3390/healthcare13091073

**Published:** 2025-05-06

**Authors:** Christina Louise Lindhardt, Maria Monberg Feenstra, Heidi Faurholt, Louise Rosenlund Andersen, Marianne Kirstine Thygesen

**Affiliations:** 1Centre for Research in Patient Communication, Odense University Hospital, 5000 Odense, Denmark; hfaurholt@sanocenter.dk; 2Department of Clinical Research, Faculty of Health Sciences, University of Southern Denmark, 5000 Odense, Denmark; maria.feenstra@rsyd.dk (M.M.F.); marianne.thygesen@live.dk (M.K.T.); 3Centre for Organisational Change in Person-Centred Healthcare, Deakin University, Geelong, VIC 3217, Australia; 4Department of Gynaecology and Obstetrics, Odense University Hospital, 5000 Odense, Denmark; louise.andersen3@rsyd.dk; 5Research Unit for General Practice, Faculty of Health Sciences, University of Southern Denmark, 5000 Odense, Denmark

**Keywords:** video consultation, outpatient, gynaecology, in situ training, patient-centred communication, technology

## Abstract

**Background/Objectives:** The successful integration of video consultations in routine hospital care requires further research. This study explores how healthcare professionals experienced and engaged with a pilot training program in video consultations (VCs), focusing on patient-centred communication and technical skills. **Methods:** A qualitative study was conducted at a gynaecological outpatient clinic in a Danish university hospital. In October 2022, healthcare professionals (n = 8) piloted a training program in VCs with patients suffering from gynaecological disorders, followed by semi-structured interviews. Our data analysis was inductive and inspired by thematic analysis, as proposed by Braun and Clarke. **Results:** Our analysis resulted in an overall theme, namely feasible, with context-dependent considerations, and followed by four other themes:, namely that (1) pre in situ training presents benefits and challenges, (2) consultation via video can be an advantage to consultations via phone or in-clinic, (3) individual planning and organising is a must, and (4) video consultation calls for new competencies. **Conclusions:** Our study indicates that a training program focusing on patient-centred communication, technical skills and in situ training with peer feedback is relevant when implementing VCs. Visual contact was an advantage of VC versus phone; however, patient triage was identified as essential when planning VCs. Overall, VCs are feasible in a gynaecological outpatient setting when their implementation is supported by an in situ training program and with ongoing technical support available.

## 1. Introduction

Video consultations (VCs) have increased, with the COVID-19 pandemic as a catalyst [1]. While digitalisation in healthcare was trending pre-COVID with telemedicine initiatives in radiology, psychiatry, and cardiology, full integration was lacking due to logistical challenges and technical support [2,3,4,5]. Furthermore, education and training have not received the necessary attention for the quality and further success of implementing telemedicine in healthcare. Remote consultations are favoured as the default mode of communication with patients to reduce the risk of infection, avoid patients travelling far to the hospital, and, thus, save time [1,2,6]. Telemedicine has been recognised as a patient-friendly and cost-effective tool since the COVID-19 pandemic [7], but VCs still need to be fully implemented and integrated into routine care [8,9,10].

A Danish national strategy has incorporated delivering hospital treatment and care in patients’ home environments to enable patients to continue their daily routines [11]. In the Region of Southern Denmark, it is an aim within the healthcare system that emails, phone or video consultations facilitate at least 30% of all consultations with patients [6]. Mit Sygehus (My Hospital), an app for smartphones or web browsers (https://mitsygehus.dk (accessed on 5 May 2025)), forms the platform for patients to participate in VCs with the hospital departments [12].

In particular, one hospital in Southern Denmark experienced increased video consultations in 2021 [11]. The recently increased quantity of remote consultations does not necessarily mean that VCs fit all patient groups [2,6,8,9], e.g., HCPs hold different opinions on whether patients with skin diseases or musculoskeletal disorders are eligible for VCs. The elderly or vulnerable individuals need more support than others to engage in VCs [2]. Nor are healthcare professionals (HCPs) now fully trained to offer their patients high-quality consultations via video [11]. Video consultations enable HCPs to obtain a visual of the patient and their home environment; the technique allows for better involvement of relatives as they can join the VC from, e.g., their workplace or home [2,10,13,14]. Barriers to VCs are known, including that HCPs can feel insecure about patient compliance, feel that their relationship with patients is weaker, and be afraid of overlooking important clinical observations when offering VCs, resulting in resistance towards telemedicine [2,8,9]. However, a scoping review of teleconsultations finds that training HCPs in VC and the accompanying technical skills is essential to ensure effective and valuable clinical interventions [2]. Structured formalised training in developing skills in telemedicine is necessary, as HCPs’ self-reported confidence in using telemedicine and objective measured performance deviate [15,16].

To strengthen HCPs competencies in VCs, a sizeable Danish university hospital has designed and established e-learning modules combined with simulation training, focusing on patient-centred communication and technical skills [6]. This study piloted the training program in situ in a gynaecological outpatient setting. The study aimed to explore patients’ and healthcare professionals’ experiences with VCs when a formalised training program supports their implementation. Secondly, we aimed to explore how HCPs’ communication techniques and self-efficacy for patient-centred communication evolved from before the training program to during and after video consultations. As education and training in implementing VC in healthcare requires further attention, knowledge regarding HCPs’ perspectives is essential to identify benefits and challenges in training VCs in gynaecological settings, A setting yet to be thoroughly researched within their sub-specialities, and the use of VCs. Therefore, this study examines and reports the qualitative findings on how HCPs experience piloting the training program in video consultations (VCs) in situ in a gynaecological outpatient clinic.

The results on the patients’ experiences are under publication. However, in summary, the patients’ experiences were that using VC was feasible and could provide easier access to healthcare.

## 2. Methods

A qualitative study design was chosen to explore and understand how HCPs experience piloting a training program in VCs. The COREQ guidelines were used to plan, structure, and report this study [17].

### 2.1. Setting and Intervention

The study was conducted at a large Danish university hospital in a gynaecological outpatient clinic. Here, patients usually participate in face-to-face consultations in-clinic or are consulted by phone by an HCP, e.g., a doctor or a nurse, within a flexible period of several hours. Patients from the whole region attend consultations at the university hospital with travel times of up to 4 h per visit. Patients participating in the in situ training of VCs suffered from cancer, endometriosis, benign gynaecological disorders or urological problems. Eligible patients for VCs were patients ≥ 18 years old who could speak Danish and did not require a physical examination. Patients without an electronic personal identifier were excluded, as access to Mit Sygehus was a requirement. Specialist doctors screened patients for eligibility, and an invitation letter was sent to patients by secure email.

The training program consisted of an e-learning program of four modules, lasting a total of 45 min, blended with 2 h of theoretical introduction to patient-centred communication in VCs [18] and the Calgary–Cambridge Guide (CCG guide) [19] (Figure 1). Furthermore, the training included in situ communicative and technical skills training with amateur actors as simulated patients. After completing the training, HCPs carried out VCs in the outpatient clinic the following day with invited patients. Video consultations were conducted by HCPs using Cisco Desk screens, facilitated through the Mit Sygehus system. A technical support team was on-site and ready to support patients and HCPs with any technical issues. All involved were provided with the support team’s contact number. During the VCs, each HCP was observed by a colleague and a researcher, who provided feedback after each VC. The complete research protocol, including planning and operational tasks, is available upon request for more details.

### 2.2. Participants

Eight HCPs were selected to participate from the clinic, due to restrictions and the need to continue to manage the outpatient clinic. The eight HCPs were included to pilot the training program in the outpatient clinic, and the group included four gynaecological doctors and four specialist nurses. Participants were all Danish speakers aged 38 to 60 who were familiar with outpatient consultations via phone and face-to-face. They all had more than 5 years of experience in the gynaecological outpatient clinic, and no or minor experience with VCs.

### 2.3. Data Collection

Semi-structured interviews with HCPs were conducted by phone or face-to-face, based on the participants’ preference, 1–2 weeks after the VCs. Interviews were carried out by an experienced interviewer and educated nurse (HF) who were unknown to the HCPs, and supported by an interview guide. The guide was developed through the current literature, focusing on the study’s aim, based on three main themes, namely (a) HCPs’ experiences of the training program, (b) HCPs’ experiences of conducting VCs with patients, and (c) HPCs’ thoughts on what may benefit the further implementation of VCs in a gynaecological outpatient setting. Questions were open-ended, allowing the HCPs to speak freely and for new themes to emerge [20]. Interviews lasted between 20 and 35 min and were audio-recorded and transcribed verbatim. All but one participant validated their interview transcript.

### 2.4. Data Analysis

Following Braun and Clarke’s approach, thematic analysis was applied to the data analysis [21]. Participants’ experiences were analysed inductively, allowing unexpected findings to emerge. The analysis was carried out by H.F. and C.L.L. and consisted of six steps, beginning with rereading the complete data material. Next, initial codes were generated, following a search for themes within the codes. Themes were reviewed as the analysis process was iterative, going back and forth, revising and adjusting themes, and forming a thematic map. Finally, themes and findings were named and reported, relating them to the aim of the study and the existing literature [21]. This safeguarded the qualitative description’s validity and the thematic analysis’s credibility [22]. Furthermore, the consistency and validity of the data were raised by confirming the findings through discussions with the co-authors of this article. Discrepancies during coding, theme development, and the interpretation of the findings were resolved through discussions with co-authors, further enhancing the study findings’ consistency and validity. Braun and Clarke argue that researchers’ subjectivity adds depth to the interpretation of data and is applied through ongoing reflective discussions during the iterative process of analysis and writing up the findings. Moreover, critical coding reviews ensured consistency, paying attention to the context and writing a logbook with considerations during the analysis and interpretation process [22] (Table 1).

### 2.5. Ethics

Telemedicine adheres to the same ethical standards as traditional medical practice, including confidentiality, consent, and the right to access medical information and treatment. Thus, the online platform brings nuances to applying the principles, such as ensuring data protection on the screen and the right to withdraw. At the start of the VC consultation, the HCPs informed the patient of the privacy of the conversation. They ensured that the patient’s autonomy was maintained, and that all information shared would be documented in the confidential patient journal.

It was emphasised to HCPs that participation in the study was voluntary, that asking elaborating questions about the research was encouraged, and that confidentiality was ensured by anonymising the data before publication. All participating healthcare professionals received oral and written information before providing their written consent, in compliance with the Helsinki Declaration, with the possibility to withdraw at any time during the study. The Danish Data Protection Agency approved the study (22/41990), and the Regional Committees on Health Research Ethics for Southern Denmark registered it with number S-20222000–103. Data were collected, stored, and published following the guidelines of Danish research ethics committees and Danish law and regulations [23].

## 3. Results

From the data analysis an overall theme emerged, namely feasible, with context-dependent considerations. This was followed by another four themes, namely tgat (1) pre-in situ training presents benefits and challenges, (2) consultation via video can be an advantage to consultations via phone or in-clinic, (3) individual planning and organising is a must, and (4) video consultation calls for new competencies (Figure 2).

### 3.1. Overall Theme: Feasible, with Context-Dependent Considerations

There was generally a favourable attitude towards VCs amongst the HCPs when piloting the training program for VCs in situ in the outpatient clinic. Nonetheless, HCPs remained reflective about the implementation of VCs and expressed a need to consider the insights of both HCPs and patients regarding VCs, as expressed in the following response:


*It is essential to let the clinicians at the outpatient clinics assess when and where they would like to implement video consultation. They are close to the patients and have a feel for who among the patients could benefit. We listen to the patients and will not just implement it everywhere.*
(Participant 1, doctor)

And:


*Video consultation must be seen from the patient’s perspective and needs. We [HCPs] must be careful not to introduce it too early and thus influence the patients to accept participation.*
(Participant 2, doctor)

Clinicians noted that experience, communication skills, and direct patient involvement were crucial to achieve meaningful consultations and possible clinical assessments throughout the VC.

### 3.2. Theme 1: Pre-In Situ Training Presents Benefits and Challenges

Our participants showed appreciation for having undertaken the theoretical parts and the training program before the in situ VC in the outpatient clinic and expressed the following:


*The interaction between lecturing and practical exercises was significant*
(Participant 5, nurse)


*It is essential to have the training program before one starts on the technical and communicative part of video consultation, not only using bedside training by peers.*
(Participant 4, nurse)

Furthermore, the mix of theory and practice, peer feedback at the end of the skills lab training, and the in situ training in the outpatient clinic felt very effective and beneficial to all participants, as expressed in the following response:


*Feedback was a gift. During the two days, there could have been even more focus on feedback on the in situ training with the patients. The feedback is where one develops oneself and learns. The combination of technical, practical training and communication, including exercises, overstepped boundaries, and one learned a lot.*
(Participant 3, doctor)

However, participants expressed concerns that only piloting the training program, the approach might have the inherent risk of leaving colleagues behind, as described in the following response:


*We need the other colleagues in the team to be in (to take the course as well). We are only a few [HCPs] in this training so far. The risk is that we forget how we did it before adequately implementing it.*
(Participant 5, nurse)

Our participants felt a need to prioritise VCs as an organisational priority if the training program should support a complete implementation.

### 3.3. Theme 2: Consultation via Video Can Be Advantageous over Consultations via Phone or In-Clinic

Our participants created an environment of thoughtfulness and curiosity in implementing VCs due to their professional excitement about reaching further out to patients and providing them with a more beneficial way to meet, as expressed in the following response:


*Video [consultation] is a handy tool for face-to-face communication. I can see the benefits that they (patients) are at home—in their safe environment, and at the same time, do not have the inconvenience of coming in [to the hospital].*
(Participant 2, doctor)


*An advantage is that when there is an infection risk, the vulnerable patients may have a good conversation in their homes without exposure.*
(Participant 4, doctor)

Moreover, participants remarked upon the increased possibilities to involve patients’ relatives during outpatient follow-up, as in the following response:


*[At video consultations], the family can be present as well.*
(Participant 1, doctor)

Our participants expected patients to benefit from video consultation in various ways. However, the HCPs also experienced some personal benefits to providing patients with consultations via video, as in the following response:


*They [patients] have not met me before, so [compared to a phone consultation], it gives a much more intimate conversation when you can see each other.*
(Participant 1, doctor)

Healthcare professionals appreciated the visuals provided by VCs as they could better assess patients through their body language and home environment. Non-verbal communication was also valued, as in the following response:


*[It gives] more to see a face and make eye contact (rather than only phone consultation). One (HCPs) can watch the atmosphere and see how they are. Further, the patient can get a feel for me as a person, and it brings “safety” into the conversation.*
(Participant 3, doctor)

In addition, a practical issue was that consultation time could be optimised through VCs, as expressed in the following response:


*A closing conversation may be done more quickly on video as the patient is not in the room.*
(Participant 2, doctor)

No patient or relative was physically in the consultation room, so they did not have to leave it, saving a small amount of time at every consultation. However, in some cases, VC timeslots were considered to be too short, as in the following response:


*I experience that it [the full consultation] takes more time with video. Thus, attention must be paid to the planning of our programs.*
(Participant 8, nurse)

Implementing VCs would require adjusting patient programs in the clinic to ensure realistic timeslots for VCs and varying agendas.

Furthermore, our HCPs felt a positive difference in the patients’ engagement when comparing video consultations to phone consultations, as expressed in the following responses:


*The patient can be [if called on the telephone] [out] playing golf or driving [or] be out for a walk or in Foetex [a supermarket].*
(Participant 1, doctor)


*My experience was that the patients were more engaged and ready for the (video) conversation than for face-to-face meetings in the clinic. They appeared more severe and alert than in face-to-face meetings.*
(Participant 7, Nurse)

However, the more non-distracted and alert patients in the VCs may be more related to the difference in the scheduling procedure than the consultation form, as the VCs were scheduled for an exact time. The phone consultations were non-specific during the daytime, fitting in between operations, consultations, and administrative work.

### 3.4. Theme 3: Individual Planning and Organisation Are a Must

Our participants argued that HCPs should plan carefully if offering video consultation in a gynaecological setting to preserve the patient’s autonomy and individual needs. This was reflected in the following responses:


*Video consultation must be seen from the patient’s perspective and needs.*
(Participant 2, doctor)


*If the outcome is patient satisfaction, we will do it.*
(Participant 4, nurse)

Our participating HCPs felt that the patients had to prioritised when determining the most suitable consultation type. The video method was only expected to be beneficial in consultations without a need for physical examination, for instance, as expressed in the following response:


*Video consultation will be perfect for rehabilitating conversations where physical examination is unnecessary. Conversations about where assessment is needed and where and when plans must be agreed upon may also be doable [in video consultation].*
(Participant 1, doctor)

Furthermore, video is suitable not for all patient consultations; limits were mentioned regarding the technique, but culture and organisational changes could also be obstacles when implementing video consultation. This viewpoint was reflected in the following response:


*It’s problematic that it [a video consultation] is so inflexible compared to a phone conversation—on an ordinary day, when I receive a cancellation, I can phone—I cannot do a video [consultation] as it must be scheduled for a specific time.*
(Participant 4, doctor)

The scheduled VC felt more rigid in a healthcare system context, where every timeslot is expected to be filled with meaningful patient-centred work.

### 3.5. Theme 4: Video Consultation Calls for New Competencies

Several HCPs mentioned that when new challenges, such as using VCs, emerge, people must pick up the competencies required to overcome them. They felt that more competencies should be developed, including the ability to identify patients suitable for a video consultation, as in the following response:


*The clinicians must evaluate where they would like to implement video (consultation). They sit with the patients, listen, and have an eye for which patients could benefit.*
(Participant 1, doctor)

Healthcare professionals who know the specific patient group recommended that they handle the triage of patients for VCs, which might save resources economically in the medium term.

However, the core competencies in effectively leading and helping a patient through a video consultation were new as well, and participants addressed the importance of ensuring ongoing support in the outpatient clinic to ensure full implementation, as in the following responses:


*There are differences in how fast one (HCP) picks it up (competencies in video consultation). Skills in technique and backup are necessary.*
(Participant 4, doctor)


*Anyway, I am not equipped for this video thing [consultation]; it would make my everyday life at work difficult, and I would be sad if I were ordered [by management] to do it.*
(Participant 4, doctor)

The competencies concerning the technical parts and the specific communication issues in VCs did not come equally easily for all HCPs. Even though much learning was achieved by piloting the training program in situ with patients, the HCPs needed a backup of resourceful colleagues who could assist with technical issues and/or guide the process. Concerns about whether managers would make video consultations mandatory among HCPs were dominant among some, as for those, it felt challenging to implement video consultations during their daily routines in the clinic. They feared the technique could disturb the relationship building process with the patient, as too much was happening at the video consultation, and it became noticeable. Nonetheless, experiences were mixed, as in the following response:


*I feel much more positive now. Testing it [video consultation] and evaluating it afterwards is beneficial. However, the slight advantage I experience from video consultation gets lost compared to the lack of flexibility and difficulty in handling the tool.*
(Participant 2, doctor)

Participants expressed the benefits of piloting a training program and VCs in the outpatient clinic and the potential challenges and concerns about the future implementation process.

## 4. Discussion

The study explored how HCPs experienced piloting the training program in VCs in situ in a gynaecological outpatient clinic. Our study provides knowledge about the early-stage implementation of VC training in specialised care.

In summary, our study found that participants were generally positive towards the training program in VCs and that implementing VCs in a gynaecological outpatient clinic may be durable when supported by a formalised training program. Although most participants were optimistic about the implementation, there were benefits, challenges, and limitations. Telemedicine may improve healthcare in several ways. First, as our participants mentioned, it can increase access to HCPs in a specialised gynaecological outpatient clinic without requiring transport. This may be advantageous for patients with limited mobility, remote living patients, and patients who find it challenging to take time off work. Furthermore, it allows patients to access a gynaecological appointment without being exposed to infections in a hospital. This was also found in other clinical areas [10,14,24,25].

The participants embraced participating in a training program in VCs based on patient-centred communication and technical skills before an in situ training day as positive and valuable. Implementing patient-centred communication and developing digital communication skills already at the undergraduate level at medical school and nursing training as part of preparing for the clinic may help prepare for communication with patients. It may also allow a smooth transition to using VC in hospitals. This study serves as a pilot and an example of how the implementation of VC can be approached in specialised care.

In our study, some HCPs felt it possible to shorten the consultation, but notably, only the minutes usually take a patient to leave the physical consultation room. Others of the HCPs found VCs more time-consuming than consultations by phone or face-to-face. This was due to the new technique that required additional focus, which was also seen by others [26]. Some discovered that HCPs using VCs have more time for patients in one working day than when using other consultation forms [1,2]. Nonetheless, the quality of the consultations is of the essence and may be explored further in combination with the time perspective of VCs.

The HCPs in our study noted that patients were more prepared for VCs than a phone call or a physical hospital meeting, which helped to set the scene for a focused and meaningful consultation, using the shared agenda setting and planning the meeting with the patient. This finding is supported by the literature on patient-centred communication training [18,19]. Involving the patient in the planning preserves the patient’s autonomy [27]. Despite the video screen, the HCPs in our study felt that they had genuine meetings with their patients. In other studies, general practitioners argue that video consultations reach patients more significantly, which might be achieved with increased quality in a single consultation [1,28]. The HCPs in our study further felt that they were more observant of the patient’s body language and other nonverbal cues, including how they spoke and took time to listen to the patient, as well as how they listened. They brought this up in relation to the actual VCs. Others found that HCPs can be more aware of enforcing their body language in the VCs than during physical consultations [9], and, furthermore, that HCPs can experience deterioration in relationships when using VCs [2]. This was not the case in our study, but body language might be highly prioritised in future training. This study did not address relatives’ participation in the video consultation, as this was not a focus. However, the literature has found that VCs give better access to patient’s relatives [14]. Our participants started offering VCs due to the hypothesis that patients might benefit from VCs, which is supported by the literature [29]. Our HCPs were motivated by both patient- and personal benefits, as well as increased patient satisfaction with VCs by engaging in virtual consultations, which has also been found by other researchers.

In our setting, doctors argued for and against VC, and they acknowledged that it would not be suitable for all types of gynaecological patients, i.e., some patients need a physical examination, as other researchers have described [2,8]. Furthermore, in other settings, HCPs could be afraid of overlooking something in a VC [9]. A systematic review indicates challenges and uncertainties in technology, as reported in the Nordic countries [30]. However, the findings from our study and those of others [17] suggest that the technological challenges could be met with technical help and peer support. Our HCPs could benefit from VCs, but there were also challenges. Several obstacles were mentioned, such as the technology and the lack of flexibility. In line with this, some of our participants described how they experienced the technology as being time-consuming and difficult. They further argued that when contacting patients, they preferred phone encounters, as these could be carried out at any time and anywhere during the day at the outpatient clinic. However, while some of our participants found benefits and challenges when using VCs, others have recently noted that the success of implementing and using VCs depends on factors, such as the HCP’s identity and how the HCP values using VCs in their daily work [31].

Our participants mentioned that VCs might enhance their relationship with gynaecological patients. However, attention to patient vulnerability and diversity among gynaecological patients is necessary [32,33,34], e.g., speaking about obesity and urogynaecology issues may be complex for the patient. The HCP’s understanding of the patient’s medical history may aid in triaging patients who may benefit from VCs versus those who prefer a face-to-face or phone consultation. A cognitive disability may challenge some patients, and the VCs might be of a shorter duration. A family member or other person might also participate to bring the patient as much benefit and support as possible [14]. This might help patients with mental issues when meeting the HCP for the first time through a video call. In these cases, the HCP may be more aware of how they use their body language, tonality, eye contact, and spoken words during the consultation [18]. VCs can also successfully be used with this category of patients if a relationship has already been established [35], and it is argued that patients with, e.g., urogynaecology problems, may benefit from access to healthcare online and, therefore, time-saving travelling to the hospital [36].

The introduction of this article mentions barriers to using VCs, and we observed that our participants also expressed that there were barriers to VCs. Although the sample size from this pilot test is small and only refers to the participants’ experiences in this study, it suggests that training HCPs in communication skills and IT techniques seemingly motivated the HCPs to be more positive concerning their further use of VCs with gynaecological patients in the outpatient clinic, thus easing the implementation of this method. Training in communication skills enhances the implementation of HCPs in the clinic [18,19,37,38].

Healthcare professionals’ resistance to VCs may play an important role, and it might be beneficial for the HCPs to be regarded as co-creators in a clinical setting [39]. In our study, we tried to meet this expectation during HCPs’ scheduling and testing. The implementation of a VC is experienced as a joint effort. It should be team-based in a clinical setting to give the best care and treatment in partnership with the patient, as described by Vanstone et al. [40]. Though our participants did not mention safety, they addressed safety concerning VCs. Svendsen et al. argue that to implement VCs, it is recommended that safety measures be thought through, such as written instructions, safety identifications, informed consent, and ensuring treatment follow-up [41]. This may be one of the ongoing issues that an overall implementation in gynaecological settings may investigate to ensure the safety of patients and HCPs and ensure successful implementation. Further guidelines could be drafted to fit this specific area.

### Strengths and Limitations

Our study holds certain limitations that should be taken into consideration. Due to this study serving as a pilot, transferability should be approached cautiously before deciding upon upscaling the intervention into other settings and patient populations. As the research was piloted in only a single gynaecological outpatient clinic and within a small team of HCPs, the transferability of the results to other settings is reduced, and various perspectives may apply in different settings and cultures. Future research should include quantitative studies encompassing larger groups of participants, more professional disciplines, specialities, and cultural settings, while clinical outcomes, such as time efficiency and the quality of the VC would be beneficial to include, together with an attention to the triage of patients eligible for VCs, which we found essential during implementation. A strength of this study was the novel approach and the intervention developed to support the implementation of VCs in hospitals that can inspire further development and implementation of VCs in other settings. In the future, VCs will provide a more significant number of patients with easier access to and more flexible healthcare.

## 5. Conclusions

Our results revealed positive experiences of piloting a training program and conducting VCs at a gynaecological outpatient clinic. VCs change the basis for assessing gynaecological patients, where parts of the consultation process, clinical experience, and patient involvement are crucial. The HCPs in this study had, through e-learning blended with in situ simulation and communicative training, adapted to the work procedure of video as a consultation option. They experienced new functions beyond the traditional role in a gynaecological setting. Combining virtual care with traditional face-to-face consultations has the potential to become an established consultation format for patients, offering patients possible benefits. Nonetheless, VCs might only be experienced as suitable and beneficial for some gynaecological outpatient encounters. Thus, the take-home message is that VCs in gynaecology are feasible and have benefits, but challenges and considerations are also present. Notably, the same HCPs can experience both benefits and challenges during VCs.

This study uniquely integrates and pilots a patient-centred communication training program and VC implementation in a gynaecological outpatient setting. It proves how education and implementation can be incorporated in an outpatient clinic, given that a physical examination is unnecessary. To our knowledge, it is one of the first approaches to facilitate VCs in a gynaecological setting, supported by novel pre-in situ training for HCPs in patient-centred communication and technical skills, providing insight into the early-stage implementation of VC training in a gynaecological outpatient setting. Despite VCs being implemented in healthcare, a program where training is facilitated in situ with actual patients is rare. As this study serves as a pilot, it provides a foundation for further research focusing on a broader validation, e.g., multicentre studies within diverse populations, clinical outcome measures, organisational framework development, including ethics, and bringing additional patient perspectives. The patient perspectives within this study are reported elsewhere but allow us to compare the healthcare professionals’ experience with the experience of the gynaecological patients. Together, these perspectives present an understanding of the feasibility of implementing VCs and how HCPs experienced in situ training in VCs, focusing on patient-centred communication and technical skills. Based on our results, we find that implementing VCs in a gynaecological outpatient clinic with patient triage is feasible when further supported by HCPs completing the training program in patient-centred communication and technical skills. Our recommendations for further research can guide new studies and enhance their potential to inform a structured, evidence-based implementation of VCs in specialised care.

## Figures and Tables

**Figure 1 healthcare-13-01073-f001:**
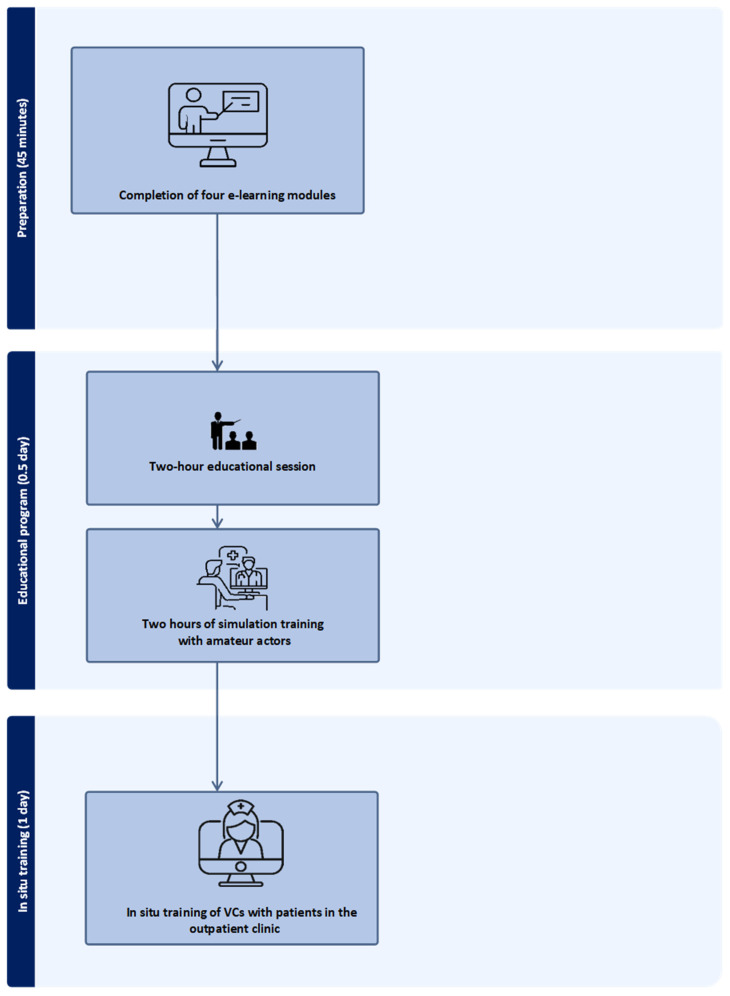
Flowchart of the intervention process and training program.

**Figure 2 healthcare-13-01073-f002:**
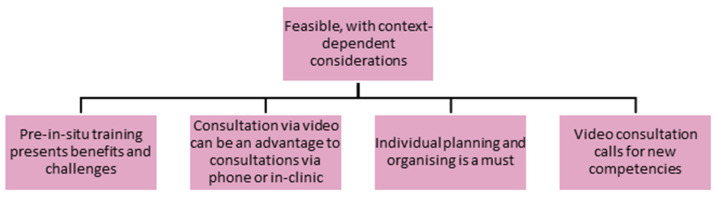
Presentation of themes.

**Table 1 healthcare-13-01073-t001:** Example of analysis process.

Theme	Category	Subcategory
Pre-in situ training presents benefits and challenges	Communication	Interaction between theory and training
	Before technical instruction	Combination of feedback and training
	New knowledge	Intercollegiate learning
	Learning skills	
	One learned a lot	
	Peer feedback is very effective	

## Data Availability

The original contributions presented in this study are included in the article. Further inquiries can be directed to the corresponding author.

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
