# Peer review of "Piloting an In Situ Training Program in Video Consultations in a Gynaecological Outpatient Clinic at a University Hospital: A Qualitative Study of the Healthcare Professionals’ Perspectives"

_healthcare, 2025, doi:10.3390/healthcare13091073_

Round 1
Reviewer 1 Report
Comments and Suggestions for Authors
Dear Authors,
Thank you for providing me with the opportunity to review this interesting paper. Below, I have listed my comments:
1) You clearly state the phenomenon (i.e. a rise in VCs due to COVID-19), but what is missing is the broader technological or systemic trends pre-COVID that set the stage (e.g., digitalisation in healthcare, pre-existing telemedicine efforts).
2) Also, more explicit framing of the problem or research gap — not just that VCs have increased, but that implementation and quality vary, and training has lagged behind, would have been helpful.
3) it is mentioned that 'VCs may not fit all patient groups', but it’s vague. Can you offer examples (e.g., elderly patients, non-native speakers, psychiatric patients)?
4) In the methods section, it would have been nice to add a few details on how discrepancies in coding among the researchers were resolved, did you reflect on your biases? how was the interview guide developed? did participants have any relevant or different level of experiences with regards to VCs?
5) In the discussion, it is best to start by restating the aim of the study. Also, remember that this is a qualitative study. Therefore, phrases like 'implementing CVs is possible' or 'Telemedicine can improve healthcare in several ways.' need to be softer as these were only the thoughts of the participants, not objective data.
I hope this feedback is helpful.
Author Response
Dear Reviewers.
Thank you for your valuable comments, which we have carefully considered. We have revised the manuscript accordingly, further language editing and final proofreading has been applied by Mr. Fred Ericks, and we hope the manuscript now meets the standards for publication.
Reviewer 1: |
Comments |
1) You clearly state the phenomenon (i.e. a rise in VCs due to COVID-19), but what is missing is the broader technological or systemic trends pre-COVID that set the stage (e.g., digitalisation in healthcare, pre-existing telemedicine efforts). |
Thank you for bringing this to our attention. We have broadened our scope in the introduction (line 38-40), added several references, and we hope you find the changes acceptable. |
2) Also, more explicit framing of the problem or research gap — not just that VCs have increased, but that implementation and quality vary, and training has lagged, would have been helpful. |
We have revised the introduction accordingly to your comment to further frame the research gap (line 38-42). We hope you find it satisfactory. |
3) it is mentioned that 'VCs may not fit all patient groups', but it’s vague. Can you offer examples (e.g., elderly patients, non-native speakers, psychiatric patients)? |
We agree upon your comment, and we have elaborated upon what patient groups might not be suitable or requiring additional support to participate in VCs (line 56-58). Hopefully, you find it acceptable. |
4) In the methods section, it would have been nice to add a few details on how discrepancies in coding among the researchers were resolved, did you reflect on your biases? how was the interview guide developed? did participants have any relevant or different level of experiences with regards to VCs? |
Thank you for focusing on enhancing the methods section and for these comments. We have now added additional information in the method section about the researcher coding progress and how potential bias was addressed, e.g., researcher subjectivity (line 147—152). Moreover, we have elaborated upon the development of the interview guide. Participants’ level of experiences with VCs are described in the Participant section. Our participants had no prior or only minor experience in VC before engaging in the training program (line123). |
5) In the discussion, it is best to start by restating the aim of the study. Also, remember that this is a qualitative study. Therefore, phrases like 'implementing CVs is possible' or 'Telemedicine can improve healthcare in several ways.' need to be softer as these were only the thoughts of the participants, not objective data. |
Thank you for raising this. As suggested, we have started the discussion section with the aim of the study (line 359-360). Moreover, we have taken attention to not generalizing results but focusing on possible ways to experience and altered the wording according to the suggestion made by you (e.g., line 364, line 366). |
Reviewer 2 Report
Comments and Suggestions for Authors
General Comments
This manuscript presents a pilot qualitative study exploring the experiences of healthcare professionals (HCPs) with an in-situ training program for video consultations (VCs) in a gynecological outpatient clinic. The study is well-structured, adheres to the COREQ guidelines, and uses Braun and Clarke’s thematic analysis appropriately. It offers practical insights into VC implementation and training design. However, there are several areas where the manuscript could be strengthened.
Strengths
The in-situ training model is highly practical and relevant to clinical settings.
The study offers a well-rounded perspective by combining theoretical training, simulation, and real patient encounters.
Emphasis on both technical and patient-centered communication skills provides a comprehensive approach to VC readiness.
Major Concerns
1. Limited Sample Size and Diversity
The sample includes only eight participants (four doctors and four nurses) from a single institution. While appropriate for a pilot study, the limited demographic and professional diversity reduces the transferability of findings.
Future research should include a broader range of disciplines, specialties, and cultural contexts.
2. Potential Bias
Some researchers were affiliated with the same institution, which may have introduced social desirability bias in participants’ responses.
The overwhelmingly positive perspectives on VC suggest possible underreporting of dissenting or ambivalent views. A more deliberate inclusion of negative or neutral opinions would enhance the study’s credibility.
3. Lack of Patient Perspectives
While the manuscript notes that patient experiences are reported elsewhere, their exclusion weakens the study’s ability to assess the full impact of VC implementation.
At minimum, a summary or reference to the patient-side findings would strengthen this paper’s conclusions.
4. Limited Clinical Outcome Data
The study focuses on HCPs’ perceptions of training, but not on clinical outcomes such as patient satisfaction, consultation quality, or time efficiency.
A future study should include quantitative outcomes to evaluate the effectiveness of VCs more robustly.
5. Ethical and Systemic Considerations Underexplored
Ethical issues such as privacy, data security, informed consent, and patient autonomy are not sufficiently addressed.
Given that some participants expressed concerns about potential pressure from management to adopt VCs, the authors should expand discussion on organizational strategy and ethical safeguards.
Minor Suggestions
Consider improving the visual presentation of the training process (e.g., a more detailed diagram or flowchart).
The expression "Possible, but do not implement everywhere" is vague. A more academic phrasing such as “Feasible with context-dependent considerations” would be clearer.
Conclusion
This study offers valuable insight into the early-stage implementation of VC training in gynecological outpatient care. As a pilot, it provides a strong foundation for further research. However, broader validation is required, including:
Multicenter studies with diverse participants
Quantitative outcome measures
Ethical and organizational framework development
Integration of patient perspectives
With these enhancements, the research has the potential to significantly inform the structured rollout of VCs in specialized care.
Author Response
Reviewer 2: |
|
General Comments Strengths The study offers a well-rounded perspective by combining theoretical training, simulation, and real patient encounters. Emphasis on both technical and patient-centered communication skills provides a comprehensive approach to VC readiness. |
Thank you for commenting on your overall impression of the manuscript. |
Major Concerns Future research should include a broader range of disciplines, specialties, and cultural contexts. |
Thank you for bringing this to our attention. We are aware of the small sample size as this was a pilot of a newly developed training program supporting the implementation of video consultations in a hospital setting. We have now addressed the limitation concerning transferability of the study results in the Strengths and Limitations section, and further broadened the discussion of future research within implementation of video consultations (line 458-465). We hope you find it acceptable.
|
2. Potential Bias The overwhelmingly positive perspectives on VC suggest possible underreporting of dissenting or ambivalent views. A more deliberate inclusion of negative or neutral opinions would enhance the study’s credibility. |
We understand your concern regarding bias in participants’ responses. We sought to minimize this bias by ensuring researchers H.F. and C.L.L. carried out all interviews as well as the analysis, as they were not affiliated with the Department of Gynaecology and Obstetrics piloting the training program but were part of a research unit at the University. M.M.F. and M.K.T. were first involved in the analysis in the discussion and confirmation of findings. Moreover, qualitative research also involves a conscious effort to identify data that contradicts or question prevailing interpretations or preconceptions of the researchers involved in the study. All to enhance the consistency and validity of findings. We find that our study results go beyond describing only positive experiences and addresses both concerns of HCPs and challenges during implementation, e.g., theme 3: addressing the logistics in VCs being a challenge in clinical practice as it has to be scheduled for a specific time (line 312-314). Moreover, theme 4: HCPs addressed a need for ongoing support because VCs required new skills not learned in a day and the lack of flexibility in VCs compared to phone consultations diminishes the advantage of VCs (line 334-339). Hopefully, you will find these quotes relevant, providing a nuanced and balanced analysis of HCPs experiences, further supporting the study’s credibility. |
3. Lack of Patient Perspectives At minimum, a summary or reference to the patient-side findings would strengthen this paper’s conclusions. |
The study reporting the patients’ experiences of piloting VCs in gynaecology outpatient clinic is accepted for publication 23.4.2025. Therefore, we are not able to provide a full summary or reference at this point. However, we have added a few remarks on the results from the patient article. We hope for your understanding as the publication process is still ongoing (line 84-87). |
4. Limited Clinical Outcome Data A future study should include quantitative outcomes to evaluate the effectiveness of VCs more robustly. |
Thank you for bringing this perspective into our considerations. We have revised the Strengths and Limitations section accordingly to your comment, and we hope you find it satisfactory (line 461-465)
|
5. Ethical and Systemic Considerations Underexplored Given that some participants expressed concerns about potential pressure from management to adopt VCs, the authors should expand discussion on organizational strategy and ethical safeguards. |
This is an important comment and in particular within the field of telemedicine, where the same ethical standards apply regarding research. We have elaborated upon this in the Ethics section, and added further information upon protecting of study participants, data handling and ensuring confidentiality. We hope our revision is acceptable (line 159-171)
|
Minor Suggestions |
We have made another suggestion for visual presentation of the intervention process and training program (Figure 1 replaced with new figure 1) illustrating the flow in the intervention set-up. We hope you find it satisfactory. |
The expression "Possible, but do not implement everywhere" is vague. A more academic phrasing such as “Feasible with context-dependent considerations” would be clearer. |
Thank you for suggesting another phrasing of the theme, which we have revised accordingly to your comment. |
Conclusion · Multicenter studies with diverse participants · Quantitative outcome measures · Ethical and organizational framework development · Integration of patient perspectives With these enhancements, the research has the potential to significantly inform the structured rollout of VCs in specialized care |
Thank you for this comment on need for a broader validation and suggestions to improve research within this field. We agree upon your comment, and we have added this in the conclusion of the manuscript. Hopefully, you will find the revision acceptable. (line 485-488), (line 489-495) |
Reviewer 3 Report
Comments and Suggestions for Authors
- This study mentioned many barriers in Introduction section, but only one sentence in the literature proves training HCPs in VC and the accompanying technical skills essential to ensure effective and valuable clinical interventions. It's not enough to support the value of your research both theoretically and practically.
- There are very big problems with this research design and results:
- Only eight healthcare professionals were selected as the sample for the study, which was a small sample size. This may lead to underrepresentation of the findings and difficulty in generalizing to a wider range of healthcare settings and populations.
- The study did not set up a control group, making it impossible to directly compare the differences between trained and untrained healthcare professionals in video consultations, and difficult to determine the actual effect of the training program.
- The research results are mainly presented in text form, lacking intuitive presentation methods such as tables and figures. It shows that the qualitative design is relatively simple, and the analysis and induction of subjects are lacking.
- From the practical and theoretical level, it is not enough to only explore healthcare professionals’ perspectives, or you need stronger theory and research design to justify this study’s perspective choice. Otherwise, it is recommended to add other more comprehensive perspectives, such as the perspective of patients.
- Although the Discussion section explains the connection with the existing research, it did not state how this research makes up for the problems in the existing research. The significance of your research is not well-reflected.
Author Response
Reviewer 3: |
|
1. This study mentioned many barriers in Introduction section, but only one sentence in the literature proves training HCPs in VC and the accompanying technical skills essential to ensure effective and valuable clinical interventions. It's not enough to support the value of your research both theoretically and practically. |
Thank you for bringing our attention to this matter. We have revised the manuscript accordingly to your comment, and elaborated upon the importance of skills training in telemedicine and provided more references to support the value of our research. We hope you find it satisfactory (line 67-69). |
2. There are very big problems with this research design and results: · Only eight healthcare professionals were selected as the sample for the study, which was a small sample size. This may lead to underrepresentation of the findings and difficulty in generalizing to a wider range of healthcare settings and populations. |
We understand your concern regarding the sample size. As mentioned under reviewer 2, we have now addressed the limitations of the study’s, transferability and wider use in research in the Strengths and Limitations section (line 458-461). Moreover, the conclusion has been revised accordingly, stressing the need for a broader validation of findings and recommendations for further research in VC training. We hope that our revisions made to the manuscript will be found acceptable by you. Furthermore, as this study serves as a pilot, a total sample of eight HCPs were the amount accepted by the Departments’ management and further being the first department to pilot the newly developed training program. Still, all eligible participants agreed to participate in the study, providing us with the study sample possible. Moreover, to put things further in perspective it was only feasible to train 8 healthcare professionals at a time, as the patient flow in the Department was unchanged during the 1,5 days allocated for piloting the VC training program. We chose to train Friday afternoon, and in situ training with patients in the gynecological outpatient clinic followed the next day (Saturday). The whole set-up required careful planning with triaging of patients, invitation letters sent to patients including a digital consent form, preparation of the rooms in the clinic with screens, planning the theoretical lecture and VC training, aligning with the IT support and making sure the HCPS were updated on start and patient flow. We therefore hope for your understanding regarding the small sample size available for this pilot study. |
· The study did not set up a control group, making it impossible to directly compare the differences between trained and untrained healthcare professionals in video consultations, and difficult to determine the actual effect of the training program. |
As this was a qualitative study to explore HCPs experiences of an in situ training program in VCs, an effect measurement was not applicable. We solely report HCPs experiences of the training program and the in situ training of VCs. We agree on your perspective that further research could apply a quantitative approach with a control group to determine the effect of the program versus not applying the program before conducting VCs in outpatient care. |
· The research results are mainly presented in text form, lacking intuitive presentation methods such as tables and figures. It shows that the qualitative design is relatively simple, and the analysis and induction of subjects are lacking. |
We appreciate your feedback. In qualitative research using Braun & Clarke it is not usual to present the results through tables and figures. However, we have in the revision of this article added Figure 2 (presentation of themes). Now Table 1 presents the analysis process and Figure 2 presents the themes (results). We have further revised Figure 1 as a flowchart to demonstrate the progress of the pilot test. We hope you find this suitable. |
4. From the practical and theoretical level, it is not enough to only explore healthcare professionals’ perspectives, or you need stronger theory and research design to justify this study’s perspective choice. Otherwise, it is recommended to add other more comprehensive perspectives, such as the perspective of patients. |
Thank you for your comment on our choice of perspective. As education and training in implementation of VC is an under-researched area in healthcare that requires further attention, we had chosen to focus the analysis, interpretation and reporting on patients’ and HCPs perspectives separately. Thus, our argumentation for taking the HCPs perspective is now incorporated in the manuscript (line 78-84). We acknowledge the importance of reporting the patients’ perspectives, which are reported in another paper, providing further knowledge and the possibility to compare the findings across the two papers. We hope our perspective and decision on reporting the perspectives of patients and HCPs separately are regarded as satisfactory by you. |
5. Although the Discussion section explains the connection with the existing research, it did not state how this research makes up for the problems in the existing research. The significance of your research is not well-reflected. |
We appreciate your comment, which we have discussed in the author group. We have rewritten sections of the article, and we have in the discussion section discussed the existing literature on barriers towards VC against the results of this research. We have revised the manuscript accordingly to your comment, and hope you find it satisfactory. (line 360-361), (line 378-381), (line 438-442). |
Round 2
Reviewer 1 Report
Comments and Suggestions for Authors
Dear Authors,
Thank you for revising the manuscript and good luck with the rest of the processes.
Author Response
Dear Reviewer 1,
Comments and Suggestions for Authors: Dear Authors, Thank you for revising the manuscript, and good luck with the rest of the process.
Author’s reply:
Thank you for reviewing our manuscript again. We appreciate your contributions in further enhancing the manuscript.
Reviewer 2 Report
Comments and Suggestions for Authors
I think it has been sufficiently corrected.
Author Response
Dear Reviewer 2,
Comments and Suggestions for Authors: It has been sufficiently corrected.
Author’s reply:
Thank you for reviewing our manuscript again. We appreciate your contributions in further enhancing the manuscript.
Reviewer 3 Report
Comments and Suggestions for Authors
Thank you very much for the author's thoughtful response.
After revision, the content of this manuscript has become richer and clearer.
However, this manuscript is more like an interview report than an academic article, as it did not provide significant differences or contributions to existing research conclusions.
From the perspective of research samples, methods, and presented results, it is still immature to analyze results solely through interviews in the current pilot context.
These findings are difficult to provide for other researchers or hospital managers to conduct similar projects because the existing experience is too messy to conclude.
Meanwhile, due to a large amount of text stacking and listing, the readability of this manuscript is very poor.
Therefore, I still do not recommend this overly hasty submission and also further publication.
Author Response
Dear Reviewer 3,
Comments and Suggestions for Authors:
Comment: Thank you very much for the author's thoughtful response.
After revision, the content of this manuscript has become richer and more precise.
However, this manuscript is more like an interview report than an academic article, as it does not provide significant differences or contributions to existing research conclusions.
Author’s reply:
I appreciate your consideration. Research that supports others’ findings is essential to publish, and our manuscript presents several findings of importance to clinical practice, as well as both benefits and challenges that require attention in further implementation of VCS. Moreover, to our knowledge, this is one of the first approaches to facilitate VC in a gynaecological setting, supported by novel pre-in situ training for HCPs in patient-centred communication and technical skills and further, providing insight into the early-stage implementation of VC training in a gynaecological outpatient setting.
Comment: From the perspective of research samples, methods, and presented results, it is still immature to analyse results solely through interviews in the current pilot context.
Author’s reply:
This qualitative study explored healthcare professionals’ experiences with piloting in situ VC training in the gynaecological outpatient setting—an under-researched area—and interviews were the most appropriate method. Before moving on to the quantitative evaluation of the program’s effect and impact, we needed to confirm that our intervention was relevant, feasible, and scalable. Pilot work in a clinical context often requires adaptations, and our design allowed us to refine the program and ensure its readiness for broader implementation (for example, in other hospital specialities). We appreciate your understanding of our stepwise, scientific approach: thoroughly developing and testing the intervention before embarking on large-scale, quantitative evaluations.
Comment: These findings are challenging for other researchers or hospital managers to conduct similar projects because the existing experience is too messy to conclude.
Author’s reply:
To support replication of our intervention, we have included a flowchart of the VC training program and its implementation process (Figure 1). We have also provided a complete description of the program in Section 2.1 (Setting and Intervention), so that others can reproduce or adapt its components. For those who would like even more detail, such as the complete research protocol, including planning and operational tasks, the authors are happy to share it upon request, which we have added in the manuscript lines 115-116.
In addition, a major Danish university hospital has already adopted our VC training program to support managers and healthcare professionals in rolling out virtual consultations across other specialities. The VC training for HCPs is currently available for clinical departments who want to ensure that the HCPs are trained towards VCs also to ensure the quality of the Vcs and the implementation process. While this pilot study offers valuable insights into an innovative approach to implementing and evaluating VCs, the program may require further refinement before wider application in different settings or cultural contexts, which is also addressed in the manuscript.
Comment: Meanwhile, due to a large amount of text stacking and listing, the readability of this manuscript is very poor.
Therefore, I still do not recommend this overly hasty submission or further publication.
Author’s reply:
Thank you for bringing this to our attention. We have further revised the manuscript, and extra proofreading has been applied. Additional adjustments in the language have been made (highlighted in blue). We hope you find our revisions acceptable. Furthermore, we hope you will appreciate the considerable effort in planning, developing, organising, studying, and reporting this work. In addition, the patients’ experiences from the piloting of in situ video consultation training for healthcare professionals are currently under publication. This will provide valuable insights into in situ VC training within an outpatient hospital setting, viewed from the patients’ perspective.